# Visual perturbation training to reduce visual dependency in Parkinson's disease: A randomized controlled trial

Remco J. Baggen[1,2], Anke Van Bladel[1,3,4], Maarten R. Prins[5], Jennifer Stappers[1], Joke Spildooren[6], Miet De Letter[3,7], Katie Bouche[1,3], Dirk Cambier[1], Leen Maes[1,8], Patrick Santens[7,9]*

1 Department of Rehabilitation Sciences, Ghent University, Ghent, Belgium, 2 Department of Human Movement Sciences, Vrije Universiteit Amsterdam, Amsterdam, The Netherlands, 3 Department of Physical and Rehabilitation Medicine, Ghent University Hospital, Ghent, Belgium, 4 Department of Rehabilitation Sciences, University of Antwerp, Antwerp, Belgium, 5 Military Rehabilitation Center Aardenburg, Doorn, The Netherlands, 6 Faculty of Rehabilitation Sciences, REVAL-Rehabilitation Research Centre, Hasselt University, Diepenbeek, Belgium, 7 Research Group BrainComm, Ghent University, Ghent, Belgium, 8 Department of Otorhinolaryngology, Ghent University Hospital, Ghent, Belgium, 9 Department of Neurology, Ghent University Hospital, Ghent, Belgium

* patrick.santens@ugent.be

## Abstract

### Introduction

Decreased gait automaticity and increased visual dependency are important contributors to falls in people with Parkinson's disease. The aim of this study was to assess if visual perturbation training during treadmill walking decreases visual dependency in people with Parkinson's disease.

### Materials and methods

For this randomized controlled trial 25 early-to-mid-stage people with Parkinson's disease (age 50-67y) without regular freezing of gait were randomly assigned to a visual perturbation group or treadmill training-only control group. Both groups trained 2 times per week for 6 weeks. Visual perturbation training consisted of self-paced treadmill walking with perturbations applied as rotations around the sagittal axis and medio-lateral translations of the virtual reality environment. The primary outcome was visual dependency. Secondary outcomes included steady-state spatiotemporal gait parameters (gait speed, step time/length/width/frequency, and cadence), as well as self-reported (near) falls.

### Results

Group x time interaction effects revealed that visual perturbation training significantly decreased visual dependency (p < 0.001) and improved temporal gait characteristics such as step time (p = 0.012), stride time (p = 0.021) and cadence (p = 0.018)

**Data availability statement:** The anonymised datasets generated and analysed for the current study are available via Zenodo: https://zenodo.org/records/17808873 under DOI https://doi.org/10.5281/zenodo.17808873.

**Funding:** This study was funded by a grant from the Flemish Parkinson League (Vlaamse Parkinson Liga) and the King Baudouin Foundation (Koning Boudewijn Stichting), grant number 2022-J1811020-226020. The funders had no role in study design, data collection and analysis, decision to publish, or preparation of the manuscript.

**Competing interests:** The authors have declared that no competing interests exist.

compared to treadmill-only controls. However, no significant effects were found for step width, step length, gait speed, and (near) falls. Improvements in visual dependency were negatively correlated to disease progression (p = 0.004). Discussion: Visual perturbation training decreases visual dependency and improves temporal gait parameters in people with Parkinson's disease. Participants in earlier disease stages appear to benefit most from visual perturbation training but additional research is needed.

## Clinical trial registration

This study was pre-registered at ClinicalTrials.gov (NCT05690308) on 09/01/2023.

---

## Introduction

Falls and fall-related injuries are a frequent and recurring problem in people with Parkinson's disease (PwPD), with about 60% of PwPD falling at least once a year and up to 39% of fallers experiencing recurrent falls [1,2]. Fall-related injuries often lead to decreased mobility, independence, and quality of life [3–5]. One important factor contributing to increased fall risk in PwPD is the reduced automaticity resulting from dopaminergic deficits in the striatum, leading to impaired balance control during routine activities such as gait [6,7]. Subsequently, PwPD become more reliant on sensory cues and exhibit increased reliance on cognitive resources for sensory-motor integration during motor tasks. Balance control depends on the weighted input of three sensory systems: the visual, vestibular, and proprioceptive systems [8,9]. However, input from these sensory systems can also be affected by changes in the striatal system related to Parkinson's disease (PD). More specifically, input from the vestibular system is reported to be affected more severely in PwPD than input from the visual and proprioceptive systems [10]. Indeed, previous studies have found alpha synuclein deposits in lower brainstem nuclei, as well as in the Deiters' neurons of the lateral vestibular nucleus that mediate postural control [11–13], indicating that impaired vestibular function might already play a role in the earliest stages of PD, even before the occurrence of typical motor symptoms associated with degeneration of dopamine-producing neurons in the substantia nigra. Subsequently, the deterioration of vestibular input may lead to relative overweighting of visual cues during perception of self-motion [14], thus increasing visual dependency. Visual dependency is defined as a "reduced ability to disregard visual cues in complex or conflicting visual environments" [8], and can be quantified by measuring changes in gait following visual cues or perturbations. Unreliable visual cues and perturbations can result in visual-vestibular mismatching of perceived and actual self-motion, which inhibits selection of appropriate motor strategies for balance control, and can potentially lead to impaired balance control increased fall risk [8]. As such, quantification of visual dependency can be used to assess the level of visual control during posture and equilibrium, and help to identify PwPD at risk [15]. Subsequently, this information may be used to tailor rehabilitation programs in

clinical practice. However, research on rehabilitation training to decrease visual dependency and its effects on gait and fall risk in PD is still limited.

A novel way to implement gait rehabilitation is through the use of virtual reality (VR) applications. The main advantage of VR rehabilitation is that it can be used to create immersive environments, enabling safe and task-specific exercises that mimic real-life and high-risk situations [16]. Task-specificity is especially relevant for PwPD as motor learning and more specifically learning of automatic responses is impaired, which results in increased reliance on feedforward systems of movement control and learning, and reduced transferability of learned motor skills [17]. A systematic review comparing VR rehabilitation with routine non-VR interventions in PwPD showed that VR training resulted in better outcomes for step length and balance performance compared to non-VR training [18]. In addition, positive effects on postural control and the limits of stability were reported in people with non-Parkinsonian vestibular dysfunction [19]. Visual perturbation training (VPT) is a form of VR rehabilitation that challenges balance control through sudden or unexpected movements of the projected environment. These events aim to simulate dizziness or disorientation after sudden head movements, which are often reported to evoke instability in PwPD [20]. By introducing unreliable visual cues to evoke visual-vestibular mismatching in the rehabilitation approach, visual dependency of PwPD can potentially be decreased, with the aim of improving the balance between inputs from the vestibular, proprioceptive and visual systems. This approach was recently developed and implemented as part of a pilot study at the Military Rehabilitation Center Aardenburg (Doorn, The Netherlands) and proved to be a safe way to decrease visual dependency in people with primary vestibular problems [21].

To investigate the potential effects of VPT training in PwPD a randomized controlled trial was designed. The main objective of this study was to assess if VPT on a treadmill improves spatiotemporal gait characteristics such as gait speed, cadence, step length and step time, and reduces visual dependency in PwPD. We hypothesized that VPT would significantly reduce visual dependency in PwPD and that no changes would be detected in a control group of PwPD performing regular treadmill training.

## Materials and methods

### Subjects

For this randomized controlled trial, people with idiopathic PD were recruited through the Neurology Department of Ghent University Hospital, various physical therapy sites around Belgium, and via the Flemish Parkinson Ligue between 09/01/2023 and 18/04/2024. Initial eligibility criteria were age (50–67 years), a clinical diagnosis of early- to moderate stage idiopathic PD (Hoehn & Yahr [H&Y] scale I-III), and the ability to walk unassisted for at least 20 minutes. Exclusion criteria were regular freezing of gait (cutoff for inclusion ≤ 1 episode per week, assessed using the New Freezing of Gait Questionnaire [NFOG-Q]), strong and unpredictable fluctuations of PD symptoms, the use of a deep brain stimulator or pump therapy, changes in dopaminergic medication status in the previous month, the presence of other neurological or musculoskeletal disorders that could affect gait stability, diagnosed vestibular disorders prior to the diagnosis of PD, previous orthopaedic surgeries of the lower limb (i.e., hip or knee prosthetics), a Montreal Cognitive Assessment (MoCA) score below 20, body weight above 120 kg (safety threshold for the equipment), and active participation in other studies. Due to the visual nature of the perturbations, candidates with severe visual impairments were also excluded. However, considering the size of the projection screen, light (PD-related) visual impairments were not deemed to hinder participation or affect outcomes. Participants were allowed to wear their daily visual aids at all times. After initial assessment of eligibility, participants were invited for testing. During the first test session, H&Y stage and MoCA score were determined for final inclusion. Following inclusion, participants were randomly assigned to the experimental visual perturbation training group (VPT) or the regular treadmill training control group (CONT) using a computer-generated four-block randomization algorithm. Participants were blinded to their group allocation. All participants were allowed to continue their regular physical therapy treatments during participation in this study. This study was approved by the Medical Ethics Committee of Ghent University Hospital (B6702022000525), in accordance with the Declaration of Helsinki and registered with ClinicalTrials.gov

(NCT05690308). All participants provided written informed consent prior to participation. The authors confirm that all ongoing and related trials for this drug/intervention are registered.

## Training protocol

After the initial assessment and testing session participants entered a four-week baseline period where they only received usual care. This baseline period allowed to control for non-training related fluctuations in visual dependency only. Subsequently, participants started their allocated training program, which consisted of two sessions per week for six weeks with at least one day of rest between sessions. The post-training measurements were performed at the start of the final training session to avoid effects of fatigue, resulting in 11 completed training session before the final assessment. A flow chart of the study is provided in Fig 1.

Training in the VPT group consisted of 20 minutes of self-paced walking on the instrumented dual-belt treadmill of the Gait Real-time Analysis Interactive Lab system (GRAIL, Motek Medical BV, Amsterdam, the Netherlands), while immersed in a 180° VR environment with the visual flow of the projected environment matched to the speed of the treadmill. The first five minutes of each training session were used to measure visual dependency (described hereafter in the measurement protocol). During the last 15 minutes visual perturbations were applied according to a fixed progression program with 12 levels (see Fig 2).

The first 4 levels included only rotations of the projected environment around the sagittal axis with increasing amplitude per successive session (at 50, 100, 150 and 200% of the amplitude applied during visual dependency test). Levels 5–8 included only medio-lateral translations with increasing amplitude (also up to 200%). Levels 9–12 combined rotational and medio-lateral translations with increasing amplitude (again up to 200%). A contingency plan was in place in case participants could not complete the training at any given perturbation intensity, however all participants were able

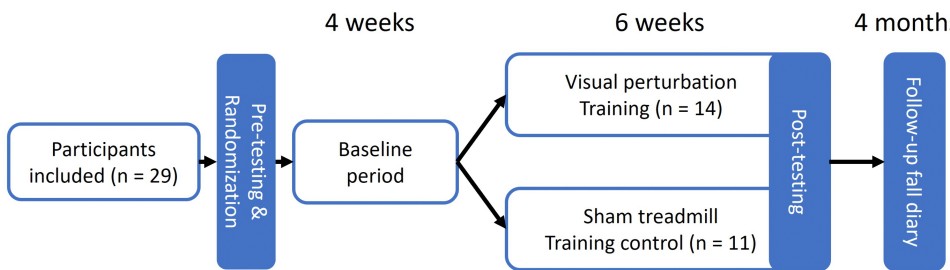

**Fig 1. Flow chart.**

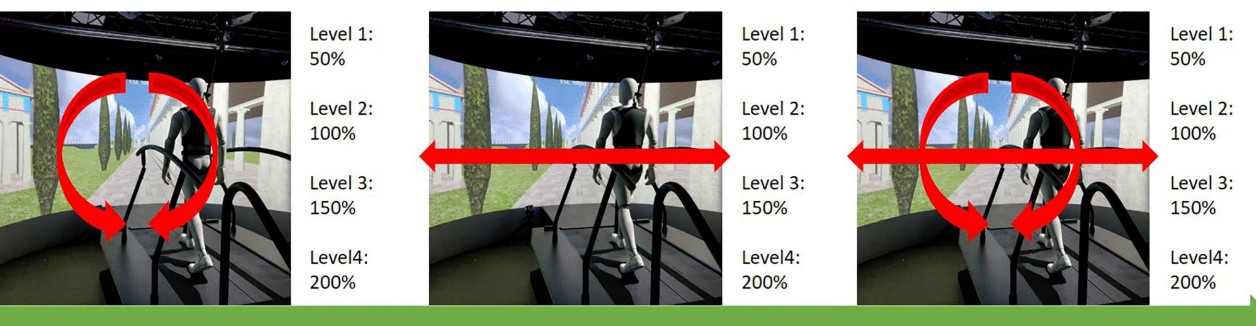

**Fig 2. Training progression scheme.**

to complete the training according to the original progression schedule. If a participant was unable to attend more than two consecutive training sessions (i.e., more than one week), participation in the study was terminated due to possible detraining effects. The CONT group received 20 minutes of treadmill training at self-selected speed without VR, visual perturbations or other visual cues. During the measurements and training sessions all participants were secured using a suspended harness, without supporting body weight, to prevent falls. Participants were not allowed to hold the handrails and were instructed to keep looking at the screen (during VR sessions) or wall (during CONT training) in front of them. None of the participants used the handrails during the measurements. Three brief touches (without grabbing onto the bars) of the handrails were tolerated during training. Gaze direction was monitored visually by the researchers. To mitigate any possible effects of acute hyper- or hypotension that can occur in PwPD, either pathologically or from medication, blood pressure was recorded three times at the start of each session (i.e., after five minutes of rest in sitting position, immediately after standing up, and after three minutes standing up). No acute hyper- or hypotension event was detected. All participants that used medication to treat PD-related symptoms were tested and trained in ON-phase of dopaminergic medication and asked to take their medication at a standard time and at least one hour prior to their visits (depending on prescription and ON-phase duration). All training and measurement sessions were administered by the same trained researcher.

## Measurement protocol

As described previously, disease severity and cognitive function were determined prior to inclusion. For disease severity, each participant's score on the H&Y scale was assessed using the International Parkinson and Movement Disorder Society sponsored revision of the Unified Parkinson's Disease Rating Scale (MDS-UPDRS) part III [22]. The MoCA 8.3 (Dutch version) was chosen to assess cognitive function because it has a higher sensitivity to detect subtle cognitive impairment in PwPD compared to the mini mental-state examination (MMSE) [23,24]. Normally a threshold of <21 points on the MoCA scale is indicative of clinical cognitive disability. However, one extra point of tolerance was allowed as some participants experienced difficulties with the visuo-spatial/executive task of the MoCA due to tremor.

After inclusion, additional information on disease status, medication status, demographic data and anthropometrics were collected. Because of the different types and dosages of dopaminergic medication, the levodopa equivalent daily dose (LEDD) was calculated [25] and used for further analyses. MDS-UPDRS parts I and IV [22] were administered to further characterize disease status. As stated in the inclusion criteria, the NFOG-Q [26] was used to determine the presence of freezing of gait (FoG) at intake, and to characterize freezing episodes. The 10-item Iconographical Falls Efficacy Scale (IconFES) [27] was used to assess fear of falling in daily life. Finally, participants were asked how often they experienced a fall or near-fall in the month prior to inclusion and were asked to complete a fall diary recording falls and near falls each month for a minimum period of four months post-intervention. Data from self-reported falls and near-falls were pooled per participant for analyses.

## Outcomes

**Gait characteristics.** During the second part of pre-testing, participants performed a standardized clinical gait analysis measuring spatiotemporal parameters on the GRAIL system. Participants were instructed to walk at a comfortable speed for 3–5 minutes to get familiarized with walking on the self-paced treadmill [28–30] in a VR 'Forest Road' environment with optical flow that matched the speed of the treadmill without any perturbations or obstacles. Once the participant indicated to have reached a comfortable gait speed and felt secure enough to walk without holding the handrails, three minutes of data were recorded in the same VR environment. Steady-state gait characteristics were subsequently calculated from ground reaction forces obtained from force sensors embedded in the treadmill (at 1000 samples/s) using the Gait Offline Analysis Tool (Motek Medical BV, Amsterdam, the Netherlands). Outcomes included in this study can be found in Table 1. This gait analysis protocol was repeated during post-testing.

**Table 1. Outcome parameters and descriptions.**

| Outcome parameter | Description |
|---|---|
| **Gait** | |
| Step length | Anteroposterior distance between point of initial contact of one foot and the successive point of initial contact of the opposite foot. |
| Stride length | Anteroposterior distance between successive points of initial contact of one foot. |
| Step width | Mediolateral distance between initial point of contact of one foot and the successive point of initial contact of the opposite foot. |
| Step time | Time between point of initial contact of one foot and the successive point of initial contact of the opposite foot. |
| Stride time | Time between successive points of initial contact of one foot. |
| Stance time | Duration of gait cycle where the foot is in contact with the ground. |
| Swing time | Duration of gait cycle where the foot is not in contact with the ground. |
| Double support time | Duration of gait cycle where both feet are in contact with the ground. |
| Variability | Standard deviation of any spatiotemporal parameter. |
| Cadence | Number of steps per unit of time. |
| Gait speed | Total distance travelled per unit of time. |
| **Visual dependency** | |
| Visual dependency - correlation* | Correlation between visual perturbations and ML-CoM movement, indicates similarity of visual perturbation and ML-CoM trajectories. |
| Visual dependency - regression | Regression coefficient between visual perturbations and ML-CoM movement, indicates level of change in ML-CoM associated with the change in visual perturbations. |

ML-CoM = mediolateral center of mass, * = primary outcome.

## Visual dependency

After collecting the steady-state gait data, participants rested for five minutes before the visual dependency test was administered. The visual dependency test was based on the 'Tricky Road' application developed at the Military Rehabilitation Center Aardenburg, Doorn, The Netherlands, for patients with a primary vestibular dysfunction. This application showed an endless road in an ancient Greek style landscape with elements such as temples, bridges and tunnels to provide reference points for the visual perturbations. Similar to the spatiotemporal gait test, participants were first allowed to familiarize themselves with the treadmill and reach their comfortable gait speed. After two minutes of steady-state walking, visual perturbations were applied for three minutes in the form of rotations of the projected environment around the sagittal axis. The perturbation pattern was designed to be unpredictable but was standardized to allow for comparisons between subjects and sessions. The angle (in degrees) of the VR environment around the sagittal axis was determined using the equation:

$$2\sin(0.75t) + 1.2\sin(1.23t) + 0.4\sin(t)$$

This procedure was repeated at the start of all VPT sessions to allow tracking of training progression. To determine visual dependency, 3D motion tracking data was collected (at 100 samples/s) using six retro-reflective markers attached around the hips at the left and right spina iliaca anterior superior and spina iliaca posterior superior, and at each heel with the 10-camera VICON system (Oxford Metrics, Oxford, UK) integrated in the GRAIL system. These data were used

to approximate position of the center of mass in mediolateral direction (ML-CoM). All VPT data were processed using custom Matlab scripts (Matlab R2023a, The MathWorks Inc., Natick, MA, USA). ML-CoM data were first filtered using a 4th order bidirectional (2x2nd order) Butterworth filter. Then, cross-correlations were calculated between screen and ML-CoM movements, to determine the lag of ML-CoM movements relative to the movements of the projected environment at which the correlation between the two signals was highest. The correlation and regression coefficient between these two signals (screen rotation and lag-corrected ML-CoM movement), were calculated as measures for visual dependency.

### Statistical analyses

A sample size calculation was conducted using G*Power 3.1.9.4 (repeated measures ANOVA, within-between interaction with two groups and two measurements). Based on a pilot study using the 'Tricky Road' application in vestibular patients [21], we assumed a large effect size for visual dependency (correlation) (Cohen's f = 0.40, α = 0.05, and Power = 0.80), resulting in a minimum number of 16 participants. Accounting for a 20% dropout rate at least 20 participants, divided equally between groups, had to be recruited. All data were first tested for normality using the Shapiro-Wilk test. Participant characteristics, gait characteristics and visual dependency scores at baseline were compared between groups using independent samples T-tests or Mann-Whitney U-tests for non-normal distributed data. Time x group (pre- vs. post-intervention, VPT vs. CONT) interaction effects were analysed using mixed ANOVAs. Data that were not normally distributed were first Log-transformed and tested for normality again before conducting the ANOVAs. When significant interaction effects were found, additional within-group post-hoc tests were performed using either paired samples t-tests or Wilcoxon signed-rank tests. Significance level for all analyses was set at p = 0.05. For visual dependency, additional analyses were performed using data from the VPT group in order to gain insights on factors that may identify responders and non-responders. To enable this, the VPT group was divided into 'responders' and 'non-responders', based on an arbitrary percentage of difference in visual dependency between the first and last measurement. If the difference was < 10% participants were classified as non-responders, otherwise they were classified as responders. In light of the relatively small within-group sample size Spearman rank-order correlation coefficients were chosen to calculate correlations for all outcome variables. Finally, to assess trends in training progression, visual dependency and group-based variances were calculated per session. Because variances of correlation coefficients are naturally reduced when approaching 1, Fisher's z transformation was performed for all correlation coefficients to enable assessment of variances based on normal distributions.

## Results

Initially, 29 participants were recruited. Four participants dropped out of the study after initial inclusion. Two participants (one VPT and one CONT) dropped out before training initiation due to injuries unrelated to the protocol. Two participants dropped out of the CONT group during training due to personal reasons and scheduling difficulties. No complications related to the training and testing protocols were reported. Twenty-five participants completed the study. An overview of participant characteristics and primary outcomes at baseline can be found in Table 2.

### Baseline comparisons

Results for the between-group comparisons at baseline can be found in Table 2. No significant differences were found for any of the participant characteristics, medication status, fear of falling, or visual dependency scores. For the spatio-temporal gait parameters, only marginally significant differences were found for stride time (p = 0.047) and swing time (p = 0.033).

Because motor symptoms in the early phases of PD predominantly occur unilaterally, additional analyses were performed comparing step length, step time, and the percentages of stance and swing time between the affected and non-affected legs at baseline. No significant differences were found between step length of the affected and non-affected leg (determined using the MDS-UPDRS part III) for either group, and step time in the CONT group. Statistically significant differences were found for step time (p = 0.007) in the VPT group, and percent stance time (p = 0.026 for CONT and p = 0.002 for VPT). However, the

**Table 2. Participant characteristics and outcomes at baseline (mean ± SD).**

| Outcome | Control | VPT | p |
|---|---|---|---|
| *Participant characteristics* | | | |
| Sex | 3F/ 8M | 2F/ 12M | – |
| Use of dopaminergic medication and LEDD (mg)‡ | 1 No/ 10 Yes<br>428 ± 201 | 0 No/ 14 Yes<br>496 ± 366 | –<br>0.603 |
| Age (years) | 61.36 ± 4.82 | 60.00 ± 4.93 | 0.495 |
| Height (cm) | 176.12 ± 9.57 | 177.66 ± 9.57 | 0.694 |
| Weight (kg) | 77.48 ± 13.00 | 85.28 ± 16.26 | 0.208 |
| H&Y | 1.64 ± 0.51 | 1.36 ± 0.63 | 0.245 |
| Baseline physical therapy (hours) | 0.68 ± 0.60 | 1.00 ± 0.90 | 0.324 |
| Baseline (near) falls previous month | 10 ± 27 | 5 ± 12 | 0.542 |
| IconFES | 16 ± 6 | 16 ± 4 | 0.924 |
| **Primary outcomes at baseline** | | | |
| Step width (m) | 0.12 ± 0.04 | 0.14 ± 0.03 | 0.113 |
| Step length (m) | 0.67 ± 0.06 | 0.68 ± 0.10 | 0.700† |
| Step time (s) | 0.54 ± 0.06 | 0.56 ± 0.04 | 0.196† |
| Stride length (m) | 1.35 ± 0.12 | 1.36 ± 0.19 | 0.681† |
| Stride time (s) | 1.05 ± 0.04 | 1.11 ± 0.09 | **0.047*** |
| Stance time (s) | 0.70 ± 0.03 | 0.74 ± 0.06 | 0.073 |
| Swing time (s) | 0.35 ± 0.02 | 0.38 ± 0.03 | **0.033*** |
| Gait speed (m/s) | 1.29 ± 0.14 | 1.23 ± 0.22 | 0.486 |
| Cadence (steps/m) | 114.23 ± 5.54 | 108.83 ± 8.64 | 0.085 |
| Gait speed variability ($\sigma$) | $7.09^{e+2} \pm 2.81^{e+2}$ | $6.00^{e+2} \pm 1.66^{e+2}$ | 0.198† |
| Step width variability ($\sigma$) | $2.82^{e+2} \pm 0.75^{e+2}$ | $2.64^{e+2} \pm 0.93^{e+2}$ | 0.746† |
| Step time variability ($\sigma$) | $11.27^{e+2} \pm 26.18^{e+2}$ | $1.71^{e+2} \pm 1.14^{e+2}$ | 0.311† |
| Step length variability ($\sigma$) | $5.64^{e+2} \pm 4.63^{e+2}$ | $3.79^{e+2} \pm 1.31^{e+2}$ | 0.465† |
| Visual dependency correlation | 0.88 ± 0.04 | 0.90 ± 0.03 | 0.285 |
| Visual dependency regression | $2.23^{e+2} \pm 0.74^{e+2}$ | $2.07^{e+2} \pm 0.53^{e+2}$ | 0.547† |

* = significant difference between groups at α = 0.05. † = result of Mann-Whitney U-test for non-normally distributed data. LEDD = Levodopa equivalent daily dosage [25].

differences were only 0.019 seconds and 1% (for both groups) respectively and deemed unlikely to be clinically relevant [31]. Therefore, no distinction was made between the affected and non-affected leg in further analyses of gait parameters.

## Gait characteristics

Gait data from one participant in the CONT group was excluded from pre-post intervention comparisons due to a corrupted file from the post-test analyses.

Group x time interaction effects revealed significant changes of almost all temporal parameters (step time, stride time, stance time, and swing time) and cadence in the VPT group compared to the CONT group (Table 3). In contrast, no differences were found for the spatial parameters (step width, step length and stride length). Gait speed showed a trend towards significance (p = 0.053). None of the variability measures revealed significant interaction effects.

Post-hoc tests revealed significant decreases in step time (Δ −0.05 s, p = 0.001, r = 0.85), stride time (Δ −0.08 s, p < 0.001, d = 0.06), stance time (Δ −0.07 s, p < 0.001, d = 0.05), swing time (Δ −0.03 s, p < 0.001, d = 0.02) and increased cadence (Δ +18 steps per minute, p < 0.001, d = 5.72) in the VPT group. Because group x time interaction for gait speed

**Table 3. Results of repeated measures ANOVA group x time interaction effects and effect sizes (mean ± SD).**

| Outcome | Control | | VPT | | Interaction | |
|---|---|---|---|---|---|---|
| | Pre | Post | Pre | Post | p | $\eta^2$ |
| Step width (m) | 0.12 ± 0.04 | 0.11 ± 0.04 | 0.14 ± 0.03 | 0.14 ± 0.03 | 0.929† | 0.000 |
| Step length (m) | 0.68 ± 0.06 | 0.70 ± 0.07 | 0.68 ± 0.10 | 0.76 ± 0.05 | 0.115† | 0.109 |
| Step time (s) | 0.54 ± 0.06 | 0.54 ± 0.08 | 0.56 ± 0.04 | 0.51 ± 0.03 | **0.012\*†** | 0.253 |
| Stride length (m) | 1.35 ± 0.13 | 1.40 ± 0.15 | 1.36 ± 0.19 | 1.51 ± 0.11 | 0.102† | 0.117 |
| Stride time (s) | 1.05 ± 0.04 | 1.03 ± 0.05 | 1.11 ± 0.09 | 1.03 ± 0.06 | **0.021\*** | 0.219 |
| Stance time (s) | 0.70 ± 0.03 | 0.68 ± 0.03 | 0.74 ± 0.06 | 0.67 ± 0.04 | **0.018\*** | 0.229 |
| Swing time (s) | 0.36 ± 0.02 | 0.35 ± 0.02 | 0.38 ± 0.03 | 0.35 ± 0.03 | **0.043\*** | 0.173 |
| Gait speed (m/s) | 1.29 ± 0.15 | 1.37 ± 0.18 | 1.23 ± 0.22 | 1.48 ± 0.13 | 0.053 | 0.160 |
| Cadence (steps/min) | 113.71 ± 5.56 | 116.07 ± 8.07 | 108.83 ± 8.64 | 117.47 ± 7.42 | **0.018\*** | 0.229 |
| Gait speed variability ($\sigma$) | $7.00^{e+2} \pm 2.94^{e+2}$ | $6.00^{e+2} \pm 2.11^{e+2}$ | $6.00^{e+2} \pm 1.66^{e+2}$ | $4.36^{e+2} \pm 1.69^{e+2}$ | 0.521† | 0.020 |
| Step time variability ($\sigma$) | $12.20^{e+2} \pm 27.41^{e+2}$ | $10.10^{e+2} \pm 22.86^{e+2}$ | $1.71^{e+2} \pm 1.14^{e+2}$ | $2.21^{e+2} \pm 3.98^{e+2}$ | 0.983† | 0.000 |
| Step width variability ($\sigma$) | $2.80^{e+2} \pm 0.79^{e+2}$ | $2.60^{e+2} \pm 0.70^{e+2}$ | $2.64^{e+2} \pm 0.93^{e+2}$ | $2.64^{e+2} \pm 0.93^{e+2}$ | 0.493† | 0.022 |
| Step length variability ($\sigma$) | $6.00^{e+2} \pm 4.71^{e+2}$ | $5.10^{e+2} \pm 7.03^{e+2}$ | $3.79^{e+2} \pm 1.31^{e+2}$ | $2.64^{e+2} \pm 0.93^{e+2}$ | 0.952† | 0.000 |
| VD correlation | 0.88 ± 0.04 | 0.88 ± 0.05 | 0.90 ± 0.03 | 0.71 ± 0.15 | **<0.001\*** | 0.411 |
| VD regression | $2.23^{e+2} \pm 0.74^{e+2}$ | $1.78^{e+2} \pm 0.43^{e+2}$ | $2.07^{e+2} \pm 0.53^{e+2}$ | $0.74^{e+2} \pm 0.47^{e+2}$ | **<0.001\*†** | 0.625 |

VD = visual dependency, * = significant group x time interaction effect at $\alpha = 0.05$. † = calculated using Log-transformed data.

showed a trend towards significance, an additional explorative post-hoc analysis was conducted, revealing a significant increase in gait speed in the VPT group ($\Delta + 0.25$ m/s, $p < 0.001$, $d = 0.21$). No significant changes were found for any outcome ($p \geq 0.076$) in the CONT group.

## Visual dependency

Group x time interaction effects revealed significant training-related changes of visual dependency in the VPT group compared to the CONT group (Table 3, $p < 0.001$ for both correlations and regression analyses). Post-hoc tests revealed significant training-related decreases in visual dependency correlations ($\Delta = -0.19$, $p < 0.001$, $d = 0.15$) and regression ($\Delta = -1.33e + 2$, $p < 0.001$, $r = 0.88$) in the VPT group, and a marginal but statistically significant decrease for visual dependency regression in the CONT group ($\Delta = -0.45e + 10$, $p = 0.01$, $r = 0.78$). Fig 3 shows the raw and filtered ML-CoM data, compared to the screen rotation data and correlations for one representative participant from the VPT group at pre- and post-test. This figure illustrates that, although there is clearly still an effect of the visual perturbation on general ML-CoM movement, decoupling is visible as the pattern and regularity of the ML-CoM become more dissimilar from pre- to post-intervention.

Although visual dependency decreased significantly in the VPT group, a trend of increased variance with training progression (Fig 4) is visible. After Fisher's z transformations were performed to obtain normally distributions of the data, although less pronounced, z-derived variances showed a similar trend, increasing from 0.17 to 0.34. This indicates that response to training was not uniform across participants. For the additional analyses to identify responders to VPT, the VPT group was divided into 'responders' (n = 8, ≥ 10% reduction in visual dependency) and 'non-responders' (n = 6, < 10% reduction in visual dependency). Response to VPT was found to be negatively correlated to H&Y scores ($\rho = -0.72$, $p = 0.004$) and IconFES scores ($\rho = -0.67$, $p = 0.009$), and positively correlated to step length variability at baseline ($\rho = 0.72$, $p = 0.003$). All participants in the VPT group with a H&Y score >1 (n = 4) were classified as a non-responder and the four highest IconFES scores were non-responders. Step length variability at baseline was higher for all responders

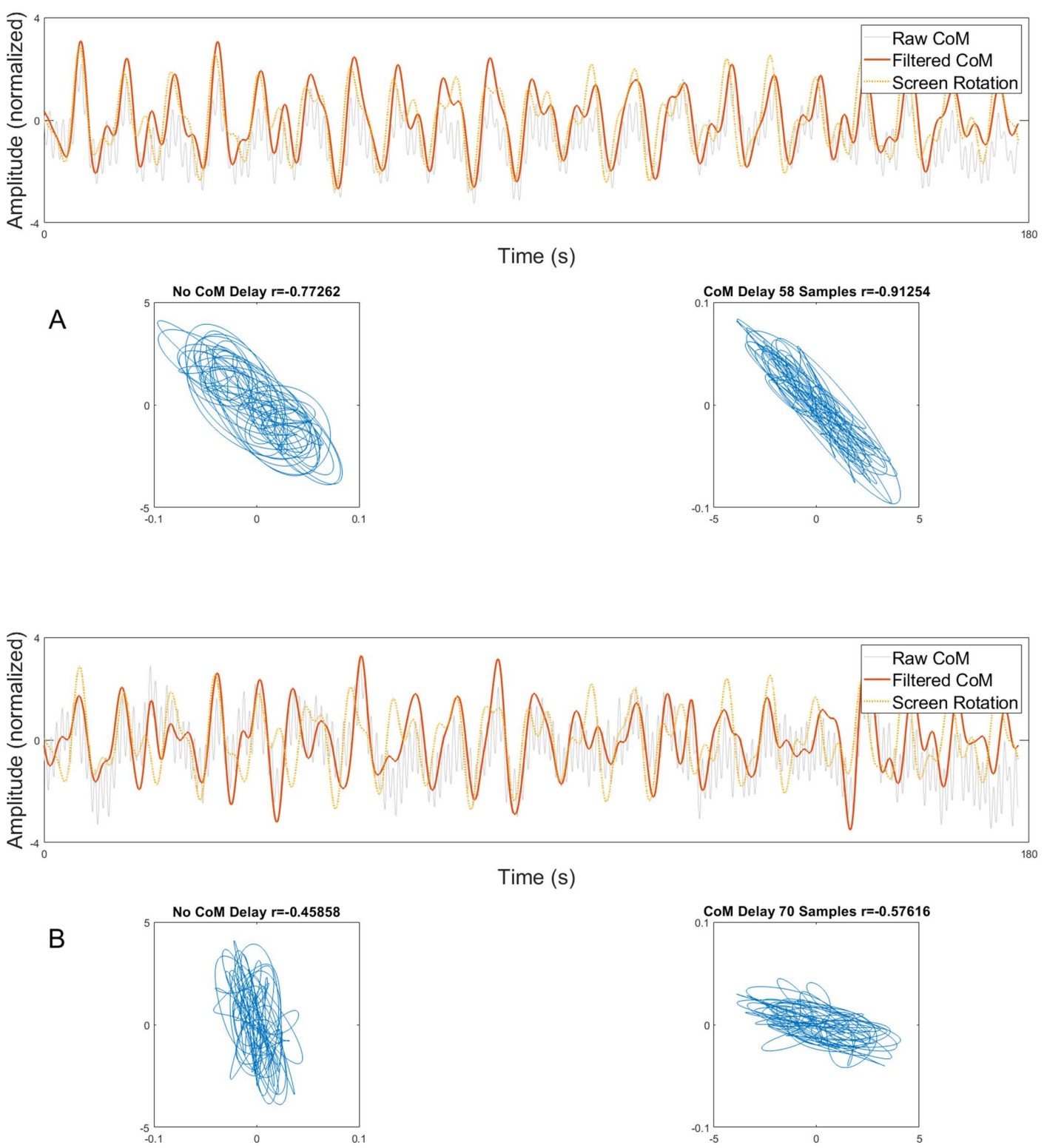

**Fig 3. Raw and filtered mediolateral center of mass (ML-CoM) data compared to screen rotations. Data of one representative participant from the VPT group pre-intervention (A) and post-intervention (B).** Correlations before and after lag correction are depicted below. Note that the filtered ML-CoM data was normalized to a mean amplitude of 1 for visualization and therefore does not depict changes in magnitude of ML-CoM movement.

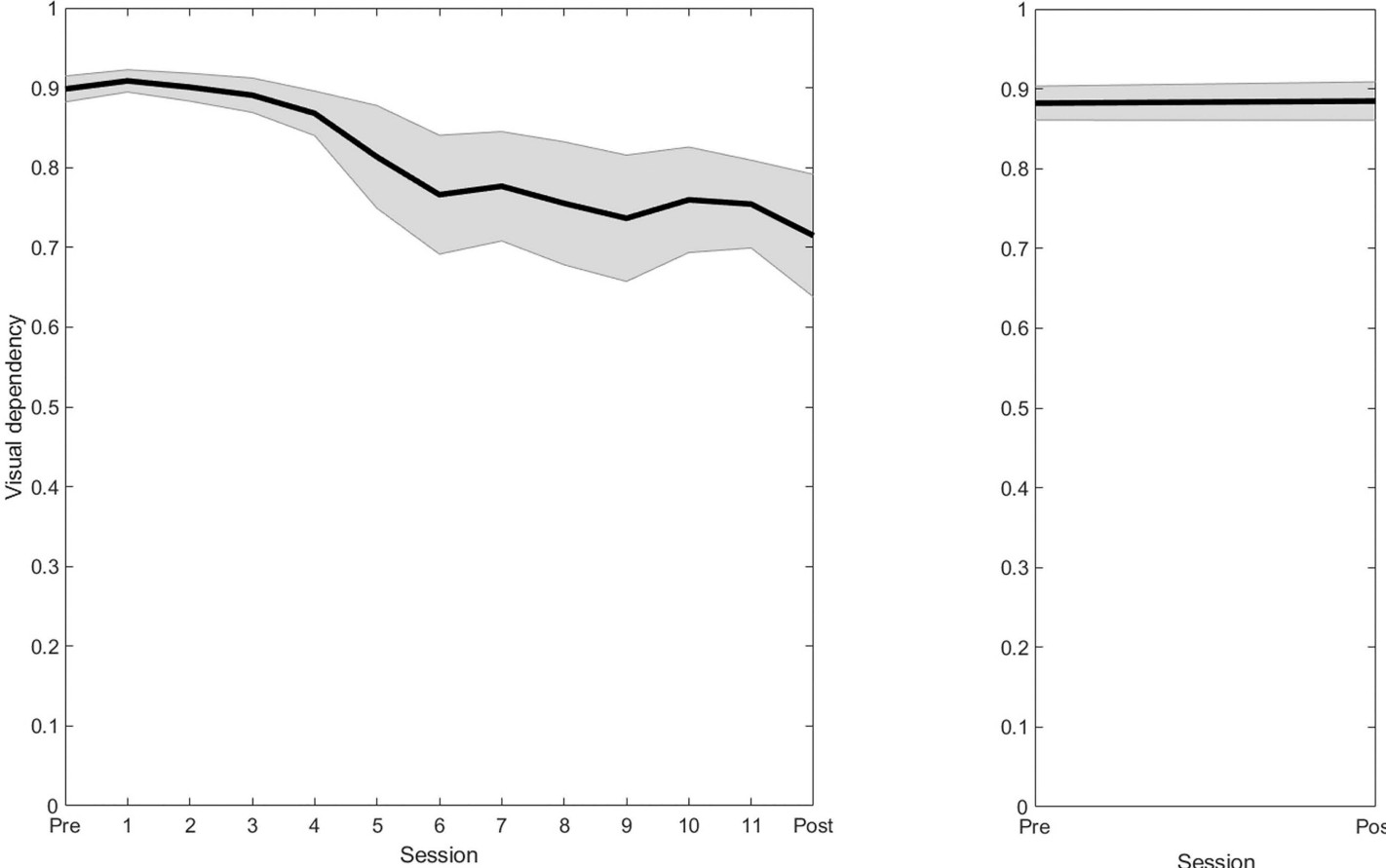

**Fig 4. Visual Dependency. Correlation between CoM and visual perturbations from all testing and training sessions for VPT (left) and pre- and post-testing from CONT (right), with standard deviations.** Values depicted from non-Fisher-transformed data.

compared to non-responders, although the size of this effect may have been amplified by results from a single responder that showed almost double the variability (0.07) of the second highest responder (0.04).

## Falls

One participant from the CONT group failed to complete the fall diaries and was excluded from further analyses. Individual data from the fall diaries during the four-month follow-up period are reported in Appendix 1. In each group one participant that reported more than 10 (near)falls per month prior to participation reported a substantial reduction in (near) falls post-intervention that remained stable up to four months follow-up. Additionally, one participant in the VPT group with more than 10 (near)falls per month before participation reported a substantial increase one month post-intervention, which decreased again and stabilized from month two. No significant within- or between-group differences were found.

## Discussion

To our knowledge, this is the first study to investigate the effects of VPT in combination with treadmill walking on visual dependency in PwPD. The main results of this single-blind randomized-controlled trial revealed that VPT can reduce visual dependency in early-stage PwPD, as well as improve temporal gait characteristics and cadence to a greater extent

than treadmill training alone. Additionally, despite the absence of a significant between-group interaction effect, a significant improvement of gait speed was found in the VPT group.

## Gait characteristics

The results from this study show that VPT can elicit greater improvements of temporal gait characteristics than conventional treadmill training alone. Similar results, such as reduced stride and swing time, have been found in other studies that employed VR treadmill training [32,33]. This could indicate decreased cognitive involvement and improved gait automaticity following VPT, allowing for faster execution of motor strategies [7]. Another possibility is that the improvements in temporal gait characteristics are attributable to exercise-related increases in neural drive [34]. However, this is less likely since such changes in neural drive are commonly associated with high intensity treadmill training and no improvements were found in the CONT group. Additionally, cadence increased significantly in the VPT group. Because increased cadence has previously been reported as a symptom of PD [35] it could be considered a negative training effect. However, increased cadence as a factor to distinguish PwPD from healthy adults was linked with decreased stride length and gait speed, factors that remained unchanged or even improved in our VPT group. Therefore, it is more likely that increased cadence in this study is a direct result of training-based improvements in stride time and gait speed.

Gait speed also improved significantly by 0.25 m/s in the VPT group. This is a promising trend as reduced gait speed is considered to be an important indicator of disease progression [36], and is in line with findings from previous treadmill-based interventions [37]. However, despite the substantial increase in gait speed, no significant between-group interaction effect was found. Several factors may account for this. First, a marginal (but non-significant) increase was found in the CONT group. This is supported by findings from a systematic review showing that treadmill training alone can have positive effects on several gait characteristics including gait speed in PwPD [38]. The statistical effects of this marginal increase in the CONT group were likely compounded by the relatively high within-group variance when using absolute gait speed, rather than relative improvements in gait speed as an outcome parameter. However, the most important factor is likely the gait speed of both groups at baseline (1.23 and 1.29 m/s respectively), which is markedly higher than previous findings that indicated 0.88 m/s during overground walking as the threshold for identifying community walkers in PwPD [39]. Our own pilot data from ten healthy age-matched participants revealed an average gait speed of 1.25 m/s, indicating that the PwPD did not yet show reduced gait speed typically associated with disease progression [36]. The relatively high gait speed at baseline most likely introduced a ceiling effect for training-based improvements of gait speed for some participants in this cohort.

Conversely, no improvements of spatial gait characteristics such as step width, step length, and stride length were found in either group. This is in contrast with findings from some previous VR treadmill training studies that found increased step length and reduced step width, attributed to more rapid swing phases as a result of more stable stance phases [18,32,33]. This implies that VPT does not affect gait and balance strategies despite a significant increase in speed of motor strategy execution. Another explanation may be found in the application of self-paced treadmill walking in this study versus more common fixed-speed treadmill walking (either at predefined comfortable pace or relative pace). However, previous research in healthy adults and stroke survivors has shown that differences in spatio-temporal gait parameters between these treadmill settings are likely not large enough to be clinically relevant [30,40].

Finally, no VPT-related changes in variability of gait parameters were found in this study. This finding was remarkable as the self-paced protocol was chosen specifically to allow for more ecological variability of spatiotemporal gait characteristics [30]. Indeed, variability of gait parameters have been reported to be affected negatively by PD [36,41,42], and treadmill perturbation training has been shown to reduce gait variability [37]. Nevertheless, the absence of significant differences in gait variability supports the theory that, despite faster motor strategy execution, the fundamental motor strategies appear to be unaffected by VPT in our cohort.

 

## Visual dependency

The results from this study confirm our hypothesis that VPT can reduce visual dependency in PwPD. Visual dependency showed a clear decrease following VPT for both correlations and regression analyses with large effect sizes ($\eta^2 > 0.14$). Fig 4 shows that the highest drop in visual dependency occurs after four training sessions. Furthermore, although response to VPT across training sessions became more variable, no clear plateau in improvement of visual dependency was visible, indicating that additional gains may be possible with continued training. Therefore, medium-to-long-term VPT appears to be an effective form of gait rehabilitation in PwPD. While this is the first study investigating visual dependency during gait in PwPD, similar effects of VR visual perturbations in healthy adults during quiet stance were reported in a recent study by Barbanchon et al. [43] who linked the improvements in visual dependency to habituation and/or a reweighting in the multisensory integration process. More specifically, they acknowledged that decreased displacement of the ML-CoM in response to visual perturbations may indicate habituation, either through decreased sensitivity to the visual stimulus or improving postural response strategies. The effects of habituation may be reflected by a reduced magnitude of ML-CoM displacement following applied perturbations. Additionally, there is increasing evidence that sensory reweighting could also play an important role [14,44]. Reflecting on their findings, Barbanchon et al. [43] hypothesized that repeated exposure to postural threats imposed by visual perturbations can lead to down-weighting of this input in favour of more reliable vestibular or proprioceptive input. In our study, visual dependency was defined based on correlations between ML-CoM displacement and the applied visual perturbation, which is not limited to average relative change in amplitude of displacement and therefore enables additional assessments of temporal components. As illustrated in Fig 3, following 11 bouts of VPT, the reduction in visual dependency appears to be not merely attributable to phase shifts in ML-CoM displacement since the shape of the time-series also shows a more irregular frequency and reduced coherence with the visual perturbation signal, especially during late adaptation. This change in ML-CoM coherence while maintaining dynamic balance in a visually challenging environment likely indicates down-weighting of unreliable visual cues following VPT, which is in line with findings from a previous study using a frequency response function and coherence obtained from a visual perturbation paradigm during stance in PwPD [44]. The ability to quantify visual dependency and elicit sensory reweighting using visual perturbations during gait may provide a valuable tool in clinical practice to determine the level of visual control exerted during posture and equilibrium, and help to reduce fall risk in PwPD.

Although the current study revealed a net positive effect on visual dependency, some participants in the VPT group exhibited little to no improvements. Therefore, additional post-hoc analyses were performed to identify factors that might mediate responses to VPT. Comparisons between responders and non-responders in the VPT group revealed that advanced disease status, higher self-reported fear of falling, and reduced step length variability at baseline were associated with limited improvements in visual dependency. These findings might indicate that VPT is likely not as effective in later-stage PD compared to early-stage PD. Future research should clarify if this method should be primarily considered for prevention, rather than a method to reverse gait deteriorations. A reduced response to VPT might be explained by the deterioration of motor learning and retention of learning effects with disease progression, which makes PwPD increasingly reliant on feedforward systems of motor learning and control [17]. As these post-hoc tests were based on a limited (sub-)sample of our cohort, future studies are necessary to confirm these hypotheses and identify PwPD who stand to gain the most benefit from VPT rehabilitation.

## Methodological considerations and limitations

All participants completed the training program without alterations to the preset progression plan. Participants indicated positive experiences with the experimental VPT, although some indicated light and brief disorientation immediately after stepping off the treadmill. However, these symptoms dissipated within a few minutes and no other short- or long-term adverse effects were reported. Some participants reported subjective improvements of stability and reduced disorientation

during sudden movements in daily life between training sessions, but these effects were not quantified as a part of this study. Nevertheless, VR VPT appears to be a safe and effective training modality for gait rehabilitation.

The results from this study did not reveal a clear effect of VPT on falls. Although the average number of (near)falls was reduced between self-reported falls from one month prior to participation and at one month follow-up, this result was not statistically significant. These results are likely skewed due to three extreme outliers, although no clear underlying factor explaining the high number of (near)falls in these participants could be identified. This may, in part, be attributable to inaccurate self-reporting [1,45] and pooling of falls and near-falls inflating the outcome for some participants. Therefore, it may be advisable to include home-based monitoring of (near)falls using camera-based monitoring or inertial sensor data, and specifically including people with a history of falls in order to achieve more objective data in future studies.

An important factor when assessing motor performance in PwPD is the effect of dopaminergic medication as it may affect both gait and balance [46,47]. In this study, all participants that used medication performed the testing and training protocols in ON-phase. This choice was made as it is most relevant to the medication state in which most PwPD perform the majority of their daily ambulatory tasks. Although group allocation was not stratified for medication status, no significant between-group differences were found at baseline. Furthermore, participants were instructed to take their medication at fixed times before testing and training sessions to ensure minimal effects of motor symptoms. Nevertheless, medication status should be taken into account when designing and administering VPT programs in future studies and clinical practice.

Despite the promising results regarding visual dependency and gait parameters, some limitations should be acknowledged. First, due to practical limitations, the assessors were not blinded to group allocation. However, the visual dependency test and steady state gait assessments, as well as the data processing pipelines were identical for all participants, fully automated, and provided objectively measured outcome parameters, subsequently limiting assessment bias. Second, this study only included early to mid-stage PwPD, and the sample size was likely too small to provide meaningful data on falls. Third, a Benjamini-Hochberg correction for multiple testing was initially applied to the outcomes of the repeated measures ANOVA. Although this did not impact visual dependency, significance of all gait adaptations during steady state walking rose above the critical level ($p = 0.05$). However, in light of the explorative nature and relatively small sample size of this study, and taking into consideration that the uncorrected gait parameters that showed significant improvement all exceeded the minimal detectable change threshold with relatively large effect sizes ($\eta^2 > 0.14$), the uncorrected results were reported instead. Furthermore, although participation in conventional physical therapy was reported to be limited, and no additional data on baseline physical activity level was collected, the average gait speed at baseline for both groups (>1.23 m/s) indicated a relatively high level of physical fitness in our cohort [39] which may not be representative of the entire PD population. As such, although a scalable progression program was put in place a-priori, and even 'non-responders' completed the entire program, the generalizability of these results towards efficacy and feasibility of VPT in more sedentary and impaired PwPD remains limited.. The high level of physical fitness in our cohort is likely attributable to a recruitment bias as participants which were previously already physically active are more likely to seek out physical intervention programs and have lowered motivational thresholds. This may also have contributed to the absence of between-group interaction effects in gait speed due to a possible ceiling effect. Nevertheless, a trend towards significant improvements of gait speed was evident. Therefore, if VPT-induced improvements in (temporal) gait parameters can be achieved even in a highly physically active PwPD, the potential training effects may be even greater in more sedentary populations. Additional tests should be included to assess if these improvements also translate into increased overground walking speed.

The population for this study was limited to PwPD without regular freezing within the age range of 50–67 years. Fifty years was set as the lower threshold in order to have a representative sample of conventional PD patients. Sixty-seven was set as the upper threshold to minimize any confounding effects of age-related vestibular problems on training-based improvements [48]. In addition, people without regular freezing were selected as freezing episodes would have reduced ML-CoM movements and therefore skewed results from the visual dependency tests. However, our results indicated that

disease progression and fear of falling are likely to be better indicators for limited VPT responsivity than age alone. Moreover, FoG episodes during gait are less common in early-to-mid-stage PwPD [49]. Therefore, it is worthwhile to explore if VPT can improve gait and balance in older PwPD, as well as freezers in future research.

Due to the novelty of these methods for testing visual dependency, no reference data for good or healthy performance, or minimal clinically important differences in PwPD are currently available. In this study, visual dependency was quantified as the correlation between visual perturbation and ML-CoM signals. Consequently, a correlation of zero would imply perfect anti-phase movement to the perturbations and thus an inverse relationship. It is reasonable to speculate that correlations around 0.5 should be considered the target for rehabilitation. However, even in healthy systems, there will always be a weighted effect of visual input on ML-CoM movement. Therefore, reference data from larger healthy cohorts is needed to assess the efficacy and minimally clinically important effect of VPT in PwPD.

Finally, no direct conclusions can be drawn about the effect of VPT on vestibular function and sensorimotor integration. The reported effects may have resulted from habituation, However, based on the proposed theories of sensory reweighting [14,43,44], it is reasonable to assume that sensory reweighting may play an important role in reducing visual dependency in PwPD. However, in order to definitively establish these effects, additional research involving vestibular testing batteries and physiological measurements of vestibular activity in PwPD is needed.

## Conclusions

These results show that the capacity to decrease visual dependency and improve gait is preserved in PwPD, and indicate that VPT can be a viable rehabilitation approach to improve fundamental sensory and motor mechanisms associated with fall risk in PwPD. VPT resulted in significant reductions of visual dependency, primarily in early-stage PwPD. Improvements of temporal gait characteristics (i.e., stride- and swing time), and gait speed, but not in spatial characteristics (i.e., stride length and step width) suggest increased automaticity without alterations to overall motor strategies. Therefore, VPT can be considered an effective form of gait rehabilitation in early-to-mid-stage PwPD. Future research should focus on exploring the direct effects of VPT on vestibular function and further explore which patients are most likely to benefit from VPT to optimize training efficacy.

## Supporting information

**S1 Table. Self-reported (near)falls.**
(DOCX)

**S2 Checklist. Study CONSORT checklist.**
(DOCX)

**S3 Protocol. Original study protocol.**
(DOCX)

## Author contributions

**Conceptualization:** Remco J. Baggen, Anke Van Bladel, Maarten R. Prins, Miet De Letter, Katie Bouche, Dirk Cambier, Leen Maes, Patrick Santens.

**Data curation:** Remco J. Baggen.

**Formal analysis:** Remco J. Baggen.

**Funding acquisition:** Remco J. Baggen, Anke Van Bladel, Miet De Letter, Katie Bouche, Dirk Cambier, Leen Maes, Patrick Santens.

**Investigation:** Remco J. Baggen, Jennifer Stappers.

**Methodology:** Remco J. Baggen, Anke Van Bladel, Maarten R. Prins, Miet De Letter, Katie Bouche, Dirk Cambier, Leen Maes, Patrick Santens.

**Project administration:** Remco J. Baggen, Jennifer Stappers, Joke Spildooren.

**Software:** Remco J. Baggen, Maarten R. Prins.

**Supervision:** Anke Van Bladel, Patrick Santens.

**Visualization:** Remco J. Baggen.

**Writing – original draft:** Remco J. Baggen.

**Writing – review & editing:** Remco J. Baggen, Anke Van Bladel, Maarten R. Prins, Joke Spildooren, Leen Maes, Patrick Santens.

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
