## [Decision Letter · Decision Letter 0]

18 Nov 2025

Dear Dr. Baggen,

Thank you for submitting your manuscript to PLOS ONE. After careful consideration, we feel that it has merit but does not fully meet PLOS ONE’s publication criteria as it currently stands. Therefore, we invite you to submit a revised version of the manuscript that addresses the points raised during the review process.

We look forward to receiving your revised manuscript.

Kind regards,

Leonardo A. Peyré-Tartaruga, Ph.D.

Academic Editor

PLOS ONE

**Journal Requirements:**

2. Thank you for submitting your clinical trial to PLOS ONE and for providing the name of the registry and the registration number. The information in the registry entry suggests that your trial was registered after patient recruitment began. PLOS ONE strongly encourages authors to register all trials before recruiting the first participant in a study.

1) your reasons for your delay in registering this study (after enrolment of participants started);

2) confirmation that all related trials are registered by stating: “The authors confirm that all ongoing and related trials for this drug/intervention are registered”.

“This study was funded by a grant from the Flemish Parkinson League (Vlaamse Parkinson Liga) and the King Baudouin Foundation (Koning Boudewijn Stichting), grant number 2022-J1811020-226020.”

4. Please note that funding information should not appear in any section or other areas of your manuscript. We will only publish funding information present in the Funding Statement section of the online submission form. Please remove any funding-related text from the manuscript.

5. In the online submission form you indicate that your data is not available for proprietary reasons and have provided a contact point for accessing this data. Please note that your current contact point is a co-author on this manuscript. According to our Data Policy, the contact point must not be an author on the manuscript and must be an institutional contact, ideally not an individual. Please revise your data statement to a non-author institutional point of contact, such as a data access or ethics committee, and send this to us via return email. Please also include contact information for the third party organization, and please include the full citation of where the data can be found.

6. We note that Figure 1 includes an image of a participant in the study.

**Additional Editor Comments** :

I strongly reccomend:improve the rationale of study showing that Individuals with visual impairment have often been observed to walk slower than individuals with unimpaired vision. This statement can be misplaced by typical low levels of physical activity and greater sedentary behavior in individuals with VI than the control population (e.g., PMID: **29614469).**small sample size (reviewer 1),steps taken to mitigate this risk potential for bias during data collection or processing (reviewer 2)consider including primary references where the gait abnormalities were observed in PD for example, (PMID: 33436993)- consider including the phatophysiological mechanism of these restrictions in PD.Particularly, I suggest trying to use one parameters very useful to analyze the functional mobility. Consider applying the rehabilitation locomotor index. To do this, you need just the walking speed and the lower limb length (great trochanter to the ground) or 0.54 of height (https://www.ncbi.nlm.nih.gov/pmc/articles/PMC2872302/)After, you need to apply these two simple equations:OWS (optimal walking speed, in m/s) = sqrt ( 0.25 x 9.81 x lower limb length (or 0.54 of height))LRI (locomotor rehabilitation index, in %) = 100 x walking speed / OWSThe message is obtain the walking speed normalized based on theory of dynamic similarities and given an index that represents how is the person is close to your more economical metabolically to his/her optimal walking speed and where the pendular mechanism is more optimized. To understand in depth, please read: http://www.clinicaltdd.com/text.asp?2016/1/2/86/184750This parameter was already used in normal (https://doi.org/10.1016/j.gaitpost.2021.09.191 ) and CHF (https://pubmed.ncbi.nlm.nih.gov/23059867/ ), Parkinson (https://pubmed.ncbi.nlm.nih.gov/26833853/) individuals.

Reviewers' comments:

Reviewer's Responses to Questions

**Comments to the Author**

1. Is the manuscript technically sound, and do the data support the conclusions?

Reviewer #1: Yes

Reviewer #2: Yes

2. Has the statistical analysis been performed appropriately and rigorously?

Reviewer #1: Yes

Reviewer #2: Yes

3. Have the authors made all data underlying the findings in their manuscript fully available?

Reviewer #1: Yes

Reviewer #2: Yes

4. Is the manuscript presented in an intelligible fashion and written in standard English?

Reviewer #1: Yes

Reviewer #2: Yes

Reviewer #1: This was a simple parallel design comparison consisting of a randomized controlled trial of early-to-mid-stage people with Parkinson’s disease randomly assigned to a visual perturbation group or treadmill training-only control group. The study was designed appropriately with sample size and power considerations. There was much information presented and the paper was well presented, statistically. There were some minor concerns as noted below.

Interaction effects were analysed using mixed ANOVAs. Tests for normality were performed appropriately as needed.. When significant interaction effects were found, additional within-group post-hoc tests were performed using either paired samples t-tests or Wilcoxon signed-rank tests. For visual dependency, additional analyses were performed using data from the VPT group in order to gain insights on factors that may identify responders and non-responders. The analyses were appropriately selected and executed for this investigation.

The total sample size required was about 20 individuals. The final sample was 25. The sample size was small and the number of endpoints, especially in the gait category, was large. Thus the issue of multiple comparisons should have entered as a possible consideration and should be included in the limitations remarks. They do note that post-hoc tests revealed significant decreases in main effects of step time, stride time, stance time and swing time and increased cadence ( all p<0.001) which, if adjusting for multiple testing, would be significant. VD regression and correlation which were primary were sufficiently discussed. Falls were summarized as best as possible. Also, the limitations were clearly presented.

Reviewer #2: This manuscript by Baggen et al. presents a well-designed and timely randomized controlled trial investigating the effects of Visual Perturbation Training (VPT) on visual dependency and gait in individuals with Parkinson's Disease (PwPD). The study addresses an important clinical problem, falls in PD, by targeting a potential underlying mechanism: increased visual dependency. The application of virtual reality for gait rehabilitation is a promising and innovative approach. The methodology is generally robust, with a clear protocol, appropriate outcome measures, and sound statistical analysis. The results are promising, showing a significant reduction in visual dependency and improvements in temporal gait parameters in the VPT group compared to treadmill training alone.

While the study is strong, there are several points that require clarification and discussion to strengthen the manuscript and its impact.

Major Comments

Sample Size and Generalizability: The study successfully recruited a sample size that met its a-priori power calculation. However, the final sample of 25 participants (14 VPT, 11 CONT) is relatively small, which limits the generalizability of the findings and the power of subgroup analyses (e.g., responder vs. non-responder). The authors correctly acknowledge this. Furthermore, the cohort was highly functional (high baseline gait speed, no regular freezers, relatively young), which may not be representative of the broader PD population at risk for falls. The discussion of a potential ceiling effect and recruitment bias is appropriate, but the authors should more explicitly state how these factors limit the generalizability of their conclusions.

Definition and Measurement of Visual Dependency: The concept of "visual dependency" is central to the study, but its operationalization could be explained more clearly for a broader audience.

The authors use the correlation and regression coefficient between screen rotation and ML-CoM movement. A high correlation indicates that the CoM movement closely follows the visual perturbation, which is interpreted as high visual dependency. However, it would be helpful to discuss what a "good" or "healthy" value for this metric is. Is the goal a correlation of zero?

The authors mention that the reduction in visual dependency post-VPT is not merely a phase shift, as the time-series shapes become more dissimilar (Fig 3). This is a crucial point supporting the sensory reweighting hypothesis. This interpretation should be emphasized and discussed in more depth in the results and discussion sections.

Clinical Significance and Falls: The primary outcome (visual dependency) is a laboratory-based measure. While it is a mechanistically interesting target, its direct clinical relevance needs further justification.

The study found no significant effect on (near) falls. The authors provide plausible explanations (self-report inaccuracies, pooling of falls/near-falls, small sample). However, the conclusion that VPT improves "fundamental sensory and motor mechanisms associated with fall risk" would be stronger if a correlation (even if not significant in this small sample) between the reduction in visual dependency and the reduction in falls was tested and reported. This would help bridge the gap between the mechanistic outcome and the clinical endpoint.

Blinding and Potential Bias: The manuscript states that participants were blinded, but assessors were not. While the outcomes are objective and automated, the potential for bias during data collection or processing cannot be entirely ruled out. The authors should briefly discuss the steps taken to mitigate this risk (e.g., standardized scripts, automated data processing pipelines) to reassure readers.

Minor Comments

Abstract: The abstract is clear but could briefly mention the specific patient population's characteristics (e.g., "early-to-mid-stage PwPD without regular freezing of gait") to provide better context.

Methods: Please clarify the duration of the "four-week baseline period." It is mentioned that this was a period of "usual care" before the 6-week training started. Was any data collected at the end of this baseline period to confirm stability, or was the pre-test done immediately before training initiation?

Results: The criteria for defining "responders" (≥10% reduction) is described as "arbitrary." While pragmatic, could this be justified by reference to a Minimal Clinically Important Difference (MCID) or similar concept, even if from a different field? Alternatively, a median split or regression-based approach could be considered, though the small sample size makes this challenging.

Discussion: The discussion is thorough. The speculation on why spatial parameters did not change (e.g., unaffected motor strategies) is interesting. It would be valuable to also briefly hypothesize why VPT would be expected to change spatial parameters, to give the reader a fuller picture of the theoretical framework.

Data Availability Statement: The current statement, "available from the corresponding author upon reasonable request," does not fully comply with PLOS ONE's data policy, which requires unconditional public availability. The authors should deposit their data in a public repository (e.g., Figshare, Dryad) and provide the DOI/accession number, or provide a robust justification for any restrictions.

In summary, this is a valuable and well-executed study that provides initial evidence for the efficacy of VPT in reducing visual dependency in PwPD. The manuscript is well-written, and the experimental design is sound. Addressing the points above, particularly regarding the clinical translation of the visual dependency metric and the generalizability of the findings, will significantly strengthen the manuscript and its contribution to the field.

**Do you want your identity to be public for this peer review?** For information about this choice, including consent withdrawal, please see our Privacy Policy

Reviewer #1: No

Reviewer #2: **Yes:** Andrea Paterno

---

## [Author Response · Author response to Decision Letter 1]

4 Dec 2025

See document: Response to reviewers

---

## [Decision Letter · Decision Letter 1]

3 Feb 2026

Visual Perturbation Training to Reduce Visual Dependency in Parkinson’s Disease: a Randomized Controlled Trial

PONE-D-25-41463R1

Dear Dr. Baggen,

We’re pleased to inform you that your manuscript has been judged scientifically suitable for publication and will be formally accepted for publication once it meets all outstanding technical requirements.

Kind regards,

Anne E. Martin

Academic Editor

PLOS One

Additional Editor Comments (optional):

Reviewers' comments:

Reviewer's Responses to Questions

**Comments to the Author**

Reviewer #1: All comments have been addressed

Reviewer #2: All comments have been addressed

2. Is the manuscript technically sound, and do the data support the conclusions?

Reviewer #1: (No Response)

Reviewer #2: Yes

3. Has the statistical analysis been performed appropriately and rigorously?

Reviewer #1: (No Response)

Reviewer #2: Yes

4. Have the authors made all data underlying the findings in their manuscript fully available?

Reviewer #1: (No Response)

Reviewer #2: Yes

5. Is the manuscript presented in an intelligible fashion and written in standard English?

Reviewer #1: (No Response)

Reviewer #2: Yes

Reviewer #1: (No Response)

Reviewer #2: The authors have adequately addressed all the comments raised during the previous round of review. The revised manuscript is technically sound, clearly written, and methodologically robust.

The randomized controlled design is appropriate for the research question, and the statistical analyses are appropriate and correctly interpreted. The conclusions are supported by the presented data.

The revised version shows improved clarity in the description of the intervention and outcome measures, as well as a more balanced interpretation of the findings and study limitations.

I have no further comments to raise.

**Do you want your identity to be public for this peer review?** For information about this choice, including consent withdrawal, please see our Privacy Policy

Reviewer #1: No

Reviewer #2: **Yes:** Andrea Paterno

---

## [Editor Report · Acceptance letter]

PONE-D-25-41463R1

PLOS One

Dear Dr. Baggen,

I'm pleased to inform you that your manuscript has been deemed suitable for publication in PLOS One. Congratulations! Your manuscript is now being handed over to our production team.

Kind regards,

on behalf of

Dr. Anne E. Martin

Academic Editor

PLOS One